# SEQUENCE-LEVEL FEATURES: HOW GRU AND LSTM CELLS CAPTURE $N$-GRAMS

## ABSTRACT

Modern recurrent neural networks (RNN) such as Gated Recurrent Units (GRU) and Long Short-term Memory (LSTM) have demonstrated impressive results on tasks involving sequential data in practice. Despite continuous efforts on interpreting their behaviors, the exact mechanism underlying their successes in capturing sequence-level information have not been thoroughly understood. In this work, we present a study on understanding the essential features captured by GRU/LSTM cells by mathematically expanding and unrolling the hidden states. Based on the expanded and unrolled hidden states, we find there was a type of sequence-level representations brought in by the gating mechanism, which enables the cells to encode sequence-level features along with token-level features. Specifically, we show that the cells would consist of such sequence-level features similar to those of $N$-grams. Based on such a finding, we also found that replacing the hidden states of the standard cells with $N$-gram representations does not necessarily degrade performance on the sentiment analysis and language modeling tasks, indicating such features may play a significant role for GRU/LSTM cells.

## 1 INTRODUCTION

Long Short-term Memory (LSTM) (Hochreiter & Schmidhuber, 1997) and Gated Recurrent Unit (GRU) (Chung et al., 2014) are widely used and investigated for tasks that involve sequential data. They are generally believed to be capable of capturing long-range dependencies while being able to alleviate gradient vanishing or explosion issues (Hochreiter & Schmidhuber, 1997; Karpathy et al., 2015; Sutskever et al., 2014). While such models were empirically shown to be successful across a range of tasks, certain fundamental questions such as "what essential features are GRU or LSTM cells able to capture?" have not yet been fully addressed. Lacking answers to them may limit our ability in designing better architectures.

One obstacle can be attributed to the non-linear activations used in the cells that prevent us from obtaining explicit closed-form expressions for hidden states. A possible solution is to expand the non-linear functions using the Taylor series (Arfken & Mullin, 1985) and represent hidden states with explicit input terms. Literally, each hidden state can be viewed as the combination of constituent terms capturing features of different levels of complexity. However, there is a prohibitively large number of polynomial terms involved and they can be difficult to manage. But it is possible that certain terms are more significant than others. Through a series of mathematical transformation, we found there were sequence-level representations in a form of matrix-vector multiplications among the expanded and unrolled hidden states of the GRU/LSTM cell. Such representations could represent sequence-level features that could theoretically be sensitive to the order of tokens and able to differ from the token-level features of its tokens as well as the sequence-level features of its sub-sequences, thus making it able to represent $N$-grams.

We assessed the significance of such sequence-level representations on sentiment analysis and language modeling tasks. We observed that the sequence-level representations derived from a GRU or LSTM cell were able to reflect desired properties on sentiment analysis tasks. Furthermore, in both the sentiment analysis and language modeling tasks, we replaced the GRU or LSTM cell with corresponding sequence-level representations (along with token-level representations) directly during training, and found that such models behaved similarly to the standard GRU or LSTM based models. This indicated that the sequence-level features might be significant for GRU or LSTM cells.

## 2 RELATED WORK

There have been plenty of prior works aiming to explain the behaviors of RNNs along with the variants. Early efforts were focused on exploring the empirical behaviors of recurrent neural networks (RNNs). Li et al. (2015) proposed a visualization approach to analyze intermediate representations of the LSTM-based models where certain interesting patterns could be observed. However, it might not be easy to extend to models with high-dimension representations. Greff et al. (2016) explored the performances of LSTM variants on representative tasks such as speech recognition, handwriting recognition, and argued that none of the proposed variants could significantly improve upon the standard LSTM architecture. Karpathy et al. (2015) studied the existence of interpretable cells that could capture long-range dependencies such as line lengths, quotes and brackets. However, those works did not involve the internal mechanism of GRUs or LSTMs.

Melis et al. (2020) and Krause et al. (2017) found that creating richer interaction between contexts and inputs on top of standard LSTMs could result in improvements. Their efforts actually pointed out the significance of rich interactions between inputs and contexts for LSTMs, but did not study what possible features such interactions could result in for good performances. Arras et al. (2017) applied an extended technique Layer-wise Relevance Propagation (LRP) to a bidirectional LSTM for sentiment analysis and produced reliable explanations of which words are responsible for attributing sentiment in individual text. Murdoch et al. (2018) leverage contextual decomposition methods to conduct analysis on the interactions of terms for LSTMS, which could produce importance scores for words, phrases and word interactions. A RNN unrolling technique was proposed by Sherstinsky (2018) based on signal processing concepts, transforming the RNN into the "Vanilla LSTM" network through a series of logical arguments, and Kanai et al. (2017) discussed the conditions that could prevent gradient explosions by looking into the dynamics of GRUs. Merrill et al. (2020) examined the properties of saturated RNNs and linked the update behaviors to weighted finite-state machines. Their ideas gave inspirations to explore internal behaviors of LSTM or GRU cells further. In this work, we sought to explore and study such significant underlying features.

## 3 MODEL DEFINITIONS

**Vanilla RNN** The representation of a vanilla RNN cell can be written as:
$$\boldsymbol{h}_t = \tanh(\boldsymbol{W}_i \boldsymbol{x}_t + \boldsymbol{W}_h \boldsymbol{h}_{t-1}), \tag{1}$$
where $\boldsymbol{h}_t \in \mathbb{R}^d$, $\boldsymbol{x}_t \in \mathbb{R}^{d_x}$ are the hidden state and input at time step $t$ respectively, $\boldsymbol{h}_{t-1}$ is the hidden state of the layer at time $(t-1)$ or the initial hidden state. $\boldsymbol{W}_i$ and $\boldsymbol{W}_h$ are weight matrices. Bias is suppressed here as well.

**GRU** The representation of a GRU cell can be written as [1]:
$$\begin{aligned}
\boldsymbol{r}_t &= \sigma(\boldsymbol{W}_{ir} \boldsymbol{x}_t + \boldsymbol{W}_{hr} \boldsymbol{h}_{t-1}), \\
\boldsymbol{z}_t &= \sigma(\boldsymbol{W}_{iz} \boldsymbol{x}_t + \boldsymbol{W}_{hz} \boldsymbol{h}_{t-1}), \\
\boldsymbol{n}_t &= \tanh(\boldsymbol{W}_{in} \boldsymbol{x}_t + \boldsymbol{r}_t \odot \boldsymbol{W}_{hn} \boldsymbol{h}_{t-1}), \\
\boldsymbol{h}_t &= (1 - \boldsymbol{z}_t) \odot \boldsymbol{n}_t + \boldsymbol{z}_t \odot \boldsymbol{h}_{t-1},
\end{aligned} \tag{2}$$
where $\boldsymbol{h}_t \in \mathbb{R}^d$, $\boldsymbol{x}_t \in \mathbb{R}^{d_x}$ are the hidden state and input at time step $t$ respectively, $\boldsymbol{h}_{t-1}$ is the hidden state of the layer at time $(t-1)$ or the initial hidden state. $\boldsymbol{r}_t \in \mathbb{R}^d$, $\boldsymbol{z}_t \in \mathbb{R}^d$, $\boldsymbol{n}_t \in \mathbb{R}^d$ are the reset, update, and the new gates respectively. $\boldsymbol{W}_{\star\star}$ refers to a weight matrix. $\sigma$ is the element-wise sigmoid function, and $\odot$ is the element-wise Hadamard product.

**LSTM** The representation of an LSTM cell can be written as:
$$\begin{aligned}
\boldsymbol{i}_t &= \sigma(\boldsymbol{W}_{ii} \boldsymbol{x}_t + \boldsymbol{W}_{hi} \boldsymbol{h}_{t-1}), \\
\boldsymbol{f}_t &= \sigma(\boldsymbol{W}_{if} \boldsymbol{x}_t + \boldsymbol{W}_{hf} \boldsymbol{h}_{t-1}), \\
\boldsymbol{o}_t &= \sigma(\boldsymbol{W}_{io} \boldsymbol{x}_t + \boldsymbol{W}_{ho} \boldsymbol{h}_{t-1}), \\
\boldsymbol{c}'_t &= \tanh(\boldsymbol{W}_{ic} \boldsymbol{x}_t + \boldsymbol{W}_{hc} \boldsymbol{h}_{t-1}), \\
\boldsymbol{c}_t &= \boldsymbol{f}_t \odot \boldsymbol{c}_{t-1} + \boldsymbol{i}_t \odot \boldsymbol{c}'_t, \\
\boldsymbol{h}_t &= \boldsymbol{o}_t \odot \tanh(\boldsymbol{c}_t),
\end{aligned} \tag{3}$$

---

[1] For brevity, we suppressed the bias for both GRU and LSTM cells here.

where $\boldsymbol{h}_t \in \mathbb{R}^d$, $\boldsymbol{x}_t \in \mathbb{R}^{d_x}$ are the hidden state and input at time step $t$ respectively, $\boldsymbol{i}_t$, $\boldsymbol{f}_t$, $\boldsymbol{o}_t \in \mathbb{R}^d$ are the input gate, forget gate, output gate respectively. $\boldsymbol{c}'_t \in \mathbb{R}^d$ is the new memory, $\boldsymbol{c}_t$ is the final memory. $\boldsymbol{W}_{\star\star}$ refers to a weight matrix.

## 4 UNROLLING RNNS

Using the Taylor series, the activations $\tanh(x)$ and $\sigma(x)$ can be expanded (at 0) as:

$$\tanh(x) = x + O(x^3) \ (|x| < \frac{\pi}{2}), \sigma(x) = \frac{1}{2} + \frac{1}{4}x + O(x^3) \ (|x| < \pi) \tag{4}$$

In this work, we do not seek to approximate the GRU or LSTM cells precisely, but to explore what features the cells could capture.

### 4.1 VANILLA RNN

We can expand the vanilla RNN hidden state using the Taylor series as:

$$\boldsymbol{h}_t = \boldsymbol{x}_t^n + \boldsymbol{W}_h \boldsymbol{h}_{t-1} + O(\boldsymbol{x}_t^n + \boldsymbol{W}_h \boldsymbol{h}_{t-1})^3, \tag{5}$$

where $\boldsymbol{x}_t^n = \boldsymbol{W}_i \boldsymbol{x}_t$. Let us unroll it as:

$$\boldsymbol{h}_t = \boldsymbol{x}_t^n + \sum_{i=1}^{t-1} \boldsymbol{W}_h^{t-i} \boldsymbol{x}_i^n + \boldsymbol{\epsilon}_r(\boldsymbol{x}_1, \boldsymbol{x}_2, ..., \boldsymbol{x}_t), \tag{6}$$

where $\boldsymbol{\epsilon}_r$ is the unrolled representation produced by higher-order terms. It can be seen that the vanilla RNN cell can capture the input information at each time step.

### 4.2 GRU

Let us write $\boldsymbol{x}_t^r = \boldsymbol{W}_{ir} \boldsymbol{x}_t$, $\boldsymbol{x}_t^z = \boldsymbol{W}_{iz} \boldsymbol{x}_t$, $\boldsymbol{x}_t^n = \boldsymbol{W}_{in} \boldsymbol{x}_t$, $\boldsymbol{h}_{t-1}^r = \boldsymbol{W}_{hr} \boldsymbol{h}_{t-1}$, $\boldsymbol{h}_{t-1}^z = \boldsymbol{W}_{hz} \boldsymbol{h}_{t-1}$, $\boldsymbol{h}_{t-1}^n = \boldsymbol{W}_{hn} \boldsymbol{h}_{t-1}$. Plugging Equation 4 in Equation 2, we can expand the hidden state at time step $t$, then combine like terms with respect to the *order* of $\boldsymbol{h}_{t-1}$ and represent them as:

$$\boldsymbol{h}_t = \underbrace{\frac{1}{2}\boldsymbol{x}_t^n - \frac{1}{4}\boldsymbol{x}_t^n \odot \boldsymbol{x}_t^z}_{zeroth-order}$$

$$+ \underbrace{\frac{1}{2}\boldsymbol{h}_{t-1} + \frac{1}{4}\boldsymbol{h}_{t-1}^n + \frac{1}{4}\boldsymbol{x}_t^z \odot \boldsymbol{h}_{t-1} - \frac{1}{4}\boldsymbol{x}_t^n \odot \boldsymbol{h}_{t-1}^z + \frac{1}{8}(\boldsymbol{x}_t^r - \boldsymbol{x}_t^z) \odot \boldsymbol{h}_{t-1}^n - \frac{1}{16}\boldsymbol{x}_t^r \odot \boldsymbol{x}_t^z \odot \boldsymbol{h}_{t-1}^n}_{first-order}$$

$$+ \underbrace{\frac{1}{4}\boldsymbol{h}_{t-1}^z \odot \boldsymbol{h}_{t-1} + \frac{1}{8}(\boldsymbol{h}_{t-1}^r - \boldsymbol{h}_{t-1}^z) \odot \boldsymbol{h}_{t-1}^n - \frac{1}{16}\boldsymbol{x}_t^r \odot \boldsymbol{h}_{t-1}^z \odot \boldsymbol{h}_{t-1}^n - \frac{1}{16}\boldsymbol{x}_t^z \odot \boldsymbol{h}_{t-1}^r \odot \boldsymbol{h}_{t-1}^n}_{second-order}$$

$$- \underbrace{\frac{1}{16}\boldsymbol{h}_{t-1}^z \odot \boldsymbol{h}_{t-1}^r \odot \boldsymbol{h}_{t-1}^n}_{third-order} + \boldsymbol{\xi}(\boldsymbol{x}_t, \boldsymbol{h}_{t-1}),$$

$$\tag{7}$$

where $\boldsymbol{\xi}(\boldsymbol{x}_t, \boldsymbol{h}_{t-1})$ refers to the higher-order terms of $\boldsymbol{x}_t$, $\boldsymbol{h}_{t-1}$ as well as their interactions.

We will focus on the terms involving zeroth-order and first-order terms of $\boldsymbol{h}_{t-1}$ and explore the features they can possibly result in. Then the hidden state at time step $t$ can be written as:

$$\boldsymbol{h}_t = \frac{1}{2}\boldsymbol{x}_t^n - \frac{1}{4}\boldsymbol{x}_t^n \odot \boldsymbol{x}_t^z$$
$$+ \frac{1}{2}\boldsymbol{h}_{t-1} + \frac{1}{4}\boldsymbol{h}_{t-1}^n + \frac{1}{4}\boldsymbol{x}_t^z \odot \boldsymbol{h}_{t-1} - \frac{1}{4}\boldsymbol{x}_t^n \odot \boldsymbol{h}_{t-1}^z + \frac{1}{8}(\boldsymbol{x}_t^r - \boldsymbol{x}_t^z - \frac{1}{2}\boldsymbol{x}_t^z \odot \boldsymbol{x}_t^r) \odot \boldsymbol{h}_{t-1}^n \tag{8}$$
$$+ \boldsymbol{\xi}'(\boldsymbol{x}_t, \boldsymbol{h}_{t-1}),$$

where $\boldsymbol{\xi}'(\boldsymbol{x}_t, \boldsymbol{h}_{t-1})$ refers to the higher-order terms of $\boldsymbol{h}_{t-1}$ plus $\boldsymbol{\xi}(\boldsymbol{x}_t, \boldsymbol{h}_{t-1})$. Note that the Hadamard products can be transformed into matrix-vector multiplications ($\boldsymbol{a} \odot \boldsymbol{b} = \text{diag}(\boldsymbol{a})\boldsymbol{b}$) and we can obtain the following :

$$
\begin{aligned}
\boldsymbol{h}_t = {}& \frac{1}{2}\boldsymbol{x}_t^n - \frac{1}{4}\boldsymbol{x}_t^n \odot \boldsymbol{x}_t^z \\
& + \left[ \frac{1}{2}\boldsymbol{I} + \frac{1}{4}\boldsymbol{W}_{hn} + \frac{1}{4}\text{diag}(\boldsymbol{x}_t^z) - \frac{1}{4}\text{diag}(\boldsymbol{x}_t^n)\boldsymbol{W}_{hz} + \frac{1}{8}\text{diag}(\boldsymbol{x}_t^r - \boldsymbol{x}_t^z - \frac{1}{2}\boldsymbol{x}_t^z \odot \boldsymbol{x}_t^r)\boldsymbol{W}_{hn} \right] \boldsymbol{h}_{t-1} \\
& + \boldsymbol{\xi}'(\boldsymbol{x}_t, \boldsymbol{h}_{t-1}).
\end{aligned}
\tag{9}
$$

For brevity, let us define two functions of $\boldsymbol{x}_t$:

$$
\begin{aligned}
\boldsymbol{g}(\boldsymbol{x}_t) ={}& \frac{1}{2}\boldsymbol{x}_t^n - \frac{1}{4}\boldsymbol{x}_t^n \odot \boldsymbol{x}_t^z, \\
\boldsymbol{A}(\boldsymbol{x}_t) ={}& \frac{1}{2}\boldsymbol{I} + \frac{1}{4}\boldsymbol{W}_{hn} + \frac{1}{4}\text{diag}(\boldsymbol{x}_t^z) - \frac{1}{4}\text{diag}(\boldsymbol{x}_t^n)\boldsymbol{W}_{hz} + \frac{1}{8}\text{diag}(\boldsymbol{x}_t^r - \boldsymbol{x}_t^z - \frac{1}{2}\boldsymbol{x}_t^z \odot \boldsymbol{x}_t^r)\boldsymbol{W}_{hn}.
\end{aligned}
\tag{10}
$$

Both $\boldsymbol{g}(\boldsymbol{x}_t)$ and $\boldsymbol{A}(\boldsymbol{x}_t)$ are only functions of $\boldsymbol{x}_t$. Then we can rewrite Equation 9 as:

$$
\boldsymbol{h}_t = \boldsymbol{g}(\boldsymbol{x}_t) + \boldsymbol{A}(\boldsymbol{x}_t)\boldsymbol{h}_{t-1} + \boldsymbol{\xi}'(\boldsymbol{x}_t, \boldsymbol{h}_{t-1}).
\tag{11}
$$

Throughout all the previous time steps (assuming the initial state are 0s), the hidden state at time step $t$ can be finally unrolled as:

$$
\begin{aligned}
\boldsymbol{h}_t ={}& \boldsymbol{g}(\boldsymbol{x}_t) + \sum_{i=1}^{t-1} \underbrace{\boldsymbol{A}(\boldsymbol{x}_t)\boldsymbol{A}(\boldsymbol{x}_{t-1})...\boldsymbol{A}(\boldsymbol{x}_{i+1})}_{\boldsymbol{M}_{(i+1):t}} \boldsymbol{g}(\boldsymbol{x}_i) + \boldsymbol{\epsilon}_g(\boldsymbol{x}_1, \boldsymbol{x}_2, ..., \boldsymbol{x}_t) \\
={}& \boldsymbol{g}(\boldsymbol{x}_t) + \sum_{i=1}^{t-1} \underbrace{\boldsymbol{M}_{(i+1):t}\boldsymbol{g}(\boldsymbol{x}_i)}_{\boldsymbol{\Phi}_{i:t}} + \boldsymbol{\epsilon}_g(\boldsymbol{x}_1, \boldsymbol{x}_2, ..., \boldsymbol{x}_t),
\end{aligned}
\tag{12}
$$

where $\boldsymbol{M}_{(i+1):t} = \prod_{k=t}^{i+1} \boldsymbol{A}(\boldsymbol{x}_k) \in \mathbb{R}^{d \times d}$ is the matrix-matrix product from time step $t$ to $i + 1$, $\boldsymbol{\epsilon}_g(\boldsymbol{x}_1, \boldsymbol{x}_2, ..., \boldsymbol{x}_t)$ are the unrolled representations from the higher-order terms in Equation 11. The function $\boldsymbol{g}(\boldsymbol{x}_t)$ solely encodes current input, namely token-level features and thus we call it *token-level representation*. The matrix-vector product $\boldsymbol{\Phi}_{i:t} = \boldsymbol{M}_{(i+1):t}\boldsymbol{g}(\boldsymbol{x}_i)$ encodes the tokens starting from time step $i$ and ending at time step $t$. If the matrices are different and not diagonal, any change of the order will result in a different product. Therefore, $\boldsymbol{\Phi}_{i:t}$ is able to capture the feature of the token sequence between time step $i$ and $t$ in an *order-sensitive* manner. We call it a *sequence-level representation*. Such representations are calculated sequentially from the left to the right through a sequence of vector/matrix multiplications, leading to features reminiscent of the classical $N$-grams commonly used in natural language processing (NLP). Let us use $\hat{\boldsymbol{h}}_t$ to denote the first two terms in Equation 12 as:

$$
\hat{\boldsymbol{h}}_t = \boldsymbol{g}(\boldsymbol{x}_t) + \sum_{i=1}^{t-1} \boldsymbol{\Phi}_{i:t}.
\tag{13}
$$

$\hat{\boldsymbol{h}}_t$ can be called as $N$-*gram representations* ($N \geq 1$). At time step $t$, it is able to encode current token input and all the token sequences starting from time step $i \in \{1, 2, ..., t - 1\}$ and ending at time step $t$ given an instance. In other words, it is a linear combination of current token-level input feature (can be understood as the *unigram* feature) and sequence-level features of all the possible $N$-grams ending at time step $t$. Bidirectional GRUs would be able to capture sequence-level features from both directions.

If we make a comparison with the unrolled vanilla RNN cell as discussed above, we can see that the sequence-level representation $\boldsymbol{A}(\boldsymbol{x}_t)\boldsymbol{A}(\boldsymbol{x}_{t-1})...\boldsymbol{A}(\boldsymbol{x}_{i+1})\boldsymbol{g}(\boldsymbol{x}_i)$ is more expressive than $\boldsymbol{W}_h^{t-i}\boldsymbol{x}_i^n$ ($i = 1, ..., t - 1$) when capturing the sequence level features. Specifically, the sequence level representation in GRU explicitly models interactions among input tokens, while capturing the useful order information conveyed by them. This may also be a reason why gating mechanism can bring in improved effectiveness over vanilla RNNs apart from alleviating the gradient vanishing or explosion problems.

### 4.3 LSTM

Let us write $\boldsymbol{x}_t^i = \boldsymbol{W}_{ii}\boldsymbol{x}_t$, $\boldsymbol{x}_t^f = \boldsymbol{W}_{if}\boldsymbol{x}_t$, $\boldsymbol{x}_t^o = \boldsymbol{W}_{io}\boldsymbol{x}_t$, $\boldsymbol{x}_t^c = \boldsymbol{W}_{ic}\boldsymbol{x}_t$ . Similarly, for an LSTM cell, we can expand the memory cell and the hidden state in a similar way. We also focus on the terms that involve the zeroth-order and first-order $\boldsymbol{c}_{t-1}$ as well as $\boldsymbol{h}_{t-1}$, and write the final memory cell and hidden state together as:

$$\begin{bmatrix} \boldsymbol{c}_t \\ \boldsymbol{h}_t \end{bmatrix} = \begin{bmatrix} \boldsymbol{g}_c(\boldsymbol{x}_t) \\ \boldsymbol{g}_h(\boldsymbol{x}_t) \end{bmatrix} + \begin{bmatrix} \boldsymbol{B}(\boldsymbol{x}_t) & \boldsymbol{D}(\boldsymbol{x}_t) \\ \boldsymbol{E}(\boldsymbol{x}_t) & \boldsymbol{F}(\boldsymbol{x}_t) \end{bmatrix} \begin{bmatrix} \boldsymbol{c}_{t-1} \\ \boldsymbol{h}_{t-1} \end{bmatrix} + \begin{bmatrix} \boldsymbol{\xi}_c'(\boldsymbol{x}_t, \boldsymbol{h}_{t-1}, \boldsymbol{c}_{t-1}) \\ \boldsymbol{\xi}_h'(\boldsymbol{x}_t, \boldsymbol{h}_{t-1}, \boldsymbol{c}_{t-1}) \end{bmatrix}. \tag{14}$$

where:

$$\boldsymbol{g}_c(\boldsymbol{x}_t) = \frac{1}{4}(\boldsymbol{x}_t^i + \boldsymbol{2}) \odot \boldsymbol{x}_t^c, \ \boldsymbol{B}(\boldsymbol{x}_t) = \frac{1}{4}\mathrm{diag}(\boldsymbol{x}_t^f + \boldsymbol{2}),$$

$$\boldsymbol{D}(\boldsymbol{x}_t) = \frac{1}{4}\mathrm{diag}(\boldsymbol{x}_t^i + \boldsymbol{2})\boldsymbol{W}_{hc} + \frac{1}{4}\mathrm{diag}(\boldsymbol{x}_t^c)\boldsymbol{W}_{hi},$$

$$\boldsymbol{g}_h(\boldsymbol{x}_t) = \frac{1}{4}(\boldsymbol{x}_t^o + \boldsymbol{2}) \odot \boldsymbol{g}_c(\boldsymbol{x}_t), \ \boldsymbol{E}(\boldsymbol{x}_t) = \frac{1}{4}\mathrm{diag}(\boldsymbol{x}_t^o + \boldsymbol{2})\boldsymbol{B}(\boldsymbol{x}_t), \tag{15}$$

$$\boldsymbol{F}(\boldsymbol{x}_t) = \frac{1}{4}\mathrm{diag}(\boldsymbol{x}_t^o + \boldsymbol{2})\boldsymbol{D}(\boldsymbol{x}_t) + \frac{1}{4}\mathrm{diag}(\boldsymbol{g}_c(\boldsymbol{x}_t))\boldsymbol{W}_{ho}.$$

$\boldsymbol{\xi}_c'(\boldsymbol{x}_t, \boldsymbol{h}_{t-1}, \boldsymbol{c}_{t-1})$ and $\boldsymbol{\xi}_h'(\boldsymbol{x}_t, \boldsymbol{h}_{t-1}, \boldsymbol{c}_{t-1})$ are the higher-order terms.

Let us use the matrix $\boldsymbol{A}(\boldsymbol{x}_t) \in \mathbb{R}^{2d \times 2d}$ to denote $\begin{bmatrix} \boldsymbol{B}(\boldsymbol{x}_t) & \boldsymbol{D}(\boldsymbol{x}_t) \\ \boldsymbol{E}(\boldsymbol{x}_t) & \boldsymbol{F}(\boldsymbol{x}_t) \end{bmatrix}$, then the equation above can be written as:

$$\begin{bmatrix} \boldsymbol{c}_t \\ \boldsymbol{h}_t \end{bmatrix} = \begin{bmatrix} \boldsymbol{g}_c(\boldsymbol{x}_t) \\ \boldsymbol{g}_h(\boldsymbol{x}_t) \end{bmatrix} + \boldsymbol{A}(\boldsymbol{x}_t) \begin{bmatrix} \boldsymbol{c}_{t-1} \\ \boldsymbol{h}_{t-1} \end{bmatrix} + \begin{bmatrix} \boldsymbol{\xi}_c'(\boldsymbol{x}_t, \boldsymbol{h}_{t-1}, \boldsymbol{c}_{t-1}) \\ \boldsymbol{\xi}_h'(\boldsymbol{x}_t, \boldsymbol{h}_{t-1}, \boldsymbol{c}_{t-1}) \end{bmatrix}. \tag{16}$$

And the final memory cell and hidden state can be unrolled as:

$$\begin{bmatrix} \boldsymbol{c}_t \\ \boldsymbol{h}_t \end{bmatrix} = \begin{bmatrix} \boldsymbol{g}_c(\boldsymbol{x}_t) \\ \boldsymbol{g}_t(\boldsymbol{x}_t) \end{bmatrix} + \sum_{i=1}^{t-1} \underbrace{\left[ \prod_{k=t}^{i+1} \boldsymbol{A}(\boldsymbol{x}_k) \right]}_{\boldsymbol{\Phi}_{i:t}} \begin{bmatrix} \boldsymbol{g}_c(\boldsymbol{x}_i) \\ \boldsymbol{g}_h(\boldsymbol{x}_i) \end{bmatrix} + \boldsymbol{\epsilon}_l(\boldsymbol{x}_1, \boldsymbol{x}_2, ..., \boldsymbol{x}_t), \tag{17}$$

where $\boldsymbol{\epsilon}_l(\boldsymbol{x}_1, \boldsymbol{x}_2, ..., \boldsymbol{x}_t)$ are the unrolled representations from the higher-order terms. The matrix-vector product $\boldsymbol{\Phi}_{i:t} \in \mathbb{R}^{2d}$ for the token sequence between time step $i$ and $t$ will be viewed as the sequence-level representation. Similar properties can be inferred. Analogously, we will use $\hat{\boldsymbol{c}}_t$ and $\hat{\boldsymbol{h}}_t$ to denote the first two terms in Equation 17 respectively.

## 5 EXPERIMENTS

We would assess the significance of the sequence-level representations on downstream tasks. For $N$-grams, negation is a common linguistic phenomenon that negates part or all of the meaning of a linguistic unit with negation words or phrases. Particularly in sentiment analysis, the polarity of certain $N$-grams can be negated by adding negation words or phrases. This task is a good testing ground for us to verify the effectiveness of the learned sequence-level features. Thus, we would like to examine whether the sequence-level representations could capture the negation information for $N$-grams, which is crucial for the sentiment analysis tasks. Language modeling tasks are often used in examining how capable an encoder is when extracting features from texts. We would use them to verify whether the sequence-level representations along with token-level representations could capture sufficient features during training and produce performances on par with standard GRU or LSTM cells. The statistics of the datasets are shown in Appendix A.1.

### 5.1 INTERPRET SEQUENCE-LEVEL REPRESENTATIONS

We first trained the model with the standard GRU or LSTM cell with *Adagrad* optimizers (Duchi et al., 2011), then used the learned parameters to calculate and examine the token-level and sequence-level features on sentiment analysis tasks. Final hidden states were used for classification. $L$-2 regularization was adopted. There were three layers in the model: an embedding layer, a GRU/LSTM layer, and fully-connected layer with a sigmoid/softmax function for binary/multi-class sentiment analysis.

### 5.1.1 POLARITY SCORE

We would like to use the metric *polarity score* (Sun & Lu, 2020) to help us understand the properties of the token-level features and sequence-level features. We followed the work of Sun & Lu (2020), and defined two types of "polarity scores" to quantify such polarity information, *token-level polarity score* and *sequence-level polarity score*. Such scores are able to capture the degree of association between a token (a sequence) and a specific label. For binary sentiment analysis, each polarity score is a scalar. For multi-class sentiment analysis, each polarity score is a vector corresponding to labels.

For a GRU cell, the two types of scores will be calculated as:

$$s_t^g = \boldsymbol{w}^\top \boldsymbol{g}(\boldsymbol{x}_t), \ s_{i:t}^{\boldsymbol{\Phi}} = \boldsymbol{w}^\top \boldsymbol{\Phi}_{i:t}. \tag{18}$$

For an LSTM cell, the sequence-level representation can be split into two parts: $\boldsymbol{\Phi}_{i:t} = [\boldsymbol{\Phi}_{i:t}^c, \boldsymbol{\Phi}_{i:t}^h]$ ($\boldsymbol{\Phi}_{i:t}^c, \boldsymbol{\Phi}_{i:t}^h \in \mathbb{R}^d$). The polarity scores will be calculated as:

$$s_t^g = \boldsymbol{w}^\top \boldsymbol{g}_h(\boldsymbol{x}_t), \ s_{i:t}^{\boldsymbol{\Phi}} = \boldsymbol{w}^\top \boldsymbol{\Phi}_{i:t}^h. \tag{19}$$

And the overall polarity score at time step $t$ can be viewed as the sum of the token-level polarity score, sequence-level polarity scores and other polarity scores:

$$s_t = \boldsymbol{w}^\top \boldsymbol{h}_t = s_t^g + \sum_{i=1}^{t-1} s_{i:t}^{\boldsymbol{\Phi}} + s_t^\epsilon. \tag{20}$$

where $\boldsymbol{w}$ is the fully-connected layer weight, $s_t^g$ is the token-level polarity score at time step $t$ and $s_{i:t}^{\boldsymbol{\Phi}}$ is the sequence-level polarity score for the sequence between time step $i$ and $t$, $s_t^\epsilon$ is the polarity score produced by $\boldsymbol{\epsilon}(\boldsymbol{x}_1, \boldsymbol{x}_2, ..., \boldsymbol{x}_t)$. For binary sentiment analysis, $\boldsymbol{w} \in \mathbb{R}^d$, $s_t^g \in \mathbb{R}$, $s_{i:t}^{\boldsymbol{\Phi}} \in \mathbb{R}$. For multi-class sentiment analysis, $\boldsymbol{w} \in \mathbb{R}^{d \times k}$, $s_t^g \in \mathbb{R}^k$, $s_{i:t}^{\boldsymbol{\Phi}} \in \mathbb{R}^k$, and $k$ is the label size. The overall polarity scores will be used to make decisions for sentiment analysis.

We examined sequence-level representations on the binary and 3-class Stanford Sentiment Treebank (SST) dataset with subphrase labels (Socher et al., 2013) respectively. Final models were selected based on validation performances with embeddings randomly initialized. The embedding size and hidden size were set as 300 and 1024 respectively.

### 5.1.2 SEPARATING PHRASES

We extracted all the short labeled phrases (2-5 words, 18490 positive/12199 negative) from the binary training set, and calculated the sum of the token-level polarity score and sequence-level polarity scores for each phrase using the first two terms in Equation 20. We call the sum *phrase polarity score*. Figure 1 shows that the two types of phrases can be generally separated by the phrase polarity scores. This set of experiments show that the learned sequence-level features can capture information useful in making discrimination between positive and negative phrases.

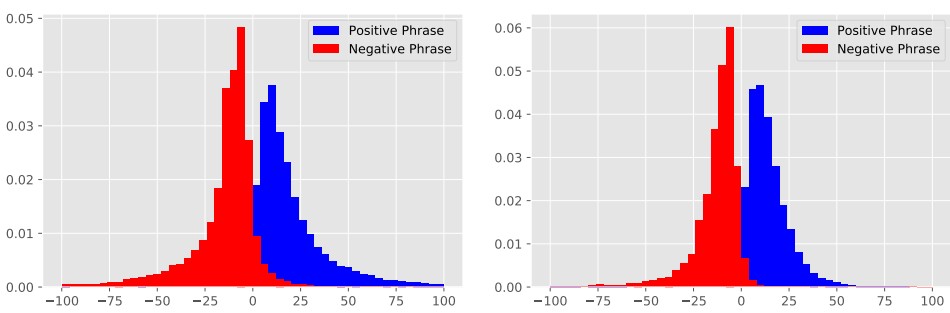

Figure 1: Phrase polarity score distribution for short phrases in binary SST. Left, GRU; right, LSTM.

### 5.1.3 NEGATING ADJECTIVES

We extracted 65 positive adjectives and 42 negative adjectives[2] following the criterion in the work of Sun & Lu (2020) from the vocabulary of the binary SST training set. We calculated the token-level polarity scores for those adjectives and sequence-level polarity scores for their corresponding negation bigrams (adding the negation word "*not*" and "*never*"). It can be seen from Table 1 that the model could likely learn to infer negations with respect to sequence-level polarity scores: the negation bigrams generally have sequence-level polarity scores of opposite signs to the corresponding adjectives. For example, "outstanding" has a large positive token-level polarity score while "not outstanding" has a large negative sequence-level polarity score.

Table 1: Statistics of token-level polarity scores for positive and negative adjectives and sequence-level polarity scores for negation bigrams. Negation words are "*not*" and "*never*". Models trained on the binary SST dataset.

| Cell | | Adjective | Polarity Score | | Remark |
|---|---|---|---|---|---|
| | | | mean | std | |
| LSTM | positive adjectives | word | 8.7 | 4.3 | token-level |
| | | negation bigram (*not* + word) | -46.5 | 13.9 | sequence-level |
| | | negation bigram (*never* + word) | -39.3 | 13.9 | sequence-level |
| | negative adjectives | word | -8.4 | 3.6 | token-level |
| | | negation bigram (*not* + word) | 14.3 | 11.8 | sequence-level |
| | | negation bigram (*never* + word) | 22.3 | 11.8 | sequence-level |
| GRU | positive adjectives | word | 8.2 | 3.8 | token-level |
| | | negation bigram (*not* + word) | -33.6 | 8.7 | sequence-level |
| | | negation bigram (*never* + word) | -27.1 | 7.9 | sequence-level |
| | negative adjectives | word | -7.7 | 3.2 | token-level |
| | | negation bigram (*not* + word) | 9.2 | 9.5 | sequence-level |
| | | negation bigram (*never* + word) | 11.0 | 8.0 | sequence-level |

### 5.1.4 DISSENTING SUB-PHRASES

We also examined whether the sequence-level representations could play a dissenting role in negating the polarity of sub-phrases. We searched for labeled phrases (3-6 tokens) that start with negation words "*not*", "*never*" and "*hardly*" and have corresponding labeled sub-phrases without negation words (those sub-phrases have opposite labels). For example, the phrase "*hardly seems worth the effort*" was labeled as "negative" while the sub-phrase "*seems worth the effort*" was labeled as "positive". Based on such conditions, we automatically extracted 14 positive phrases and 36 negative phrases along with their corresponding sub-phrases, then we calculated the polarity scores with pretrained models. We would like to see if the polarity scores assigned to such linguistic units by our models are consistent with the labels.

Table 2: Dissenting Sub-phrases: "Polarity score" refers to the sequence-level polarity scores for the longest $N$-gram in the phrases. ">0" and "<0"refers to the number of positive polarity scores and number of negative polarity scores respectively. "$ Ph" refers to the number of phrases. "Example" refers to the example phrase (with sequence-level polarity score) for each type of extracted phrases. The sub-phrases have opposite labels to the phrases. Models trained on the binary SST dataset.

| | Type | # Ph | Polarity score | | Example | |
|---|---|---|---|---|---|---|
| | | | >0 | <0 | Phrase | Polarity score |
| LSTM | positive phrases | 14 | 7 | 7 | never becomes claustrophobic | 32.3 |
| | negative phrases | 36 | 0 | 36 | hardly an objective documentary | -115.7 |
| GRU | positive phrases | 14 | 8 | 6 | never becomes claustrophobic | 10.8 |
| | negative phrases | 36 | 0 | 36 | hardly an objective documentary | -48.1 |

Ideally, based on our analysis the longest $N$-grams for the phrases will be assigned polarity scores consist with their labels to offset the impact of their sub-phrases. Table 2 shows that the sequence-level representations of the $N$-grams could be generally assigned polarity scores with the signs opposite to the sub-phrases, likely playing a dissenting role. Figure 2 shows the sequence-level

---

[2]The adjectives are listed in Table 7 in the appendix.

polarity score of the four-gram "*hardly an objective documentary*" was negatively large that could help reverse the polarity of the sub-phrase "*an objective documentary*" and make the overall polarity of the phrase negative. Such negation can be also observed on models with bidirectional GRU or LSTM cells in A.2.

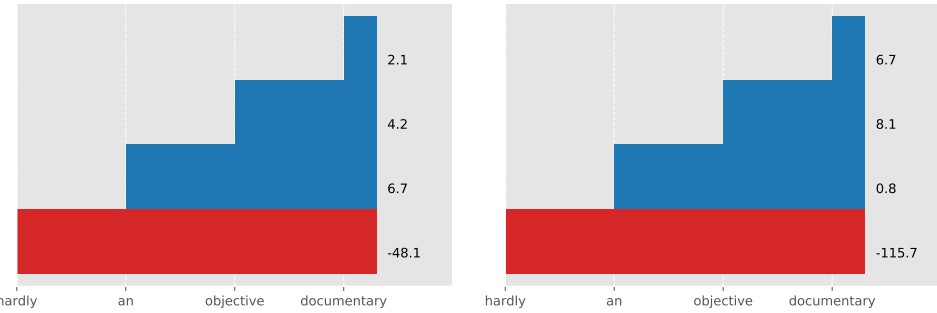

Figure 2: Example of dissenting a sub-phrase. Polarity scores are listed for $N$-grams in the phrase "*hardly an objective documentary*". Each vertical bar represents either a token or a sequence that starts from its left side and ends with its right side. Left, GRU; right, LSTM.

We noticed that in order for such cells as GRU/LSTM to learn complex compositional language structures, it may be essential for the model to have enough exposure to relevant structures during the training phase. We did a simple controlled experiment on two sets of labeled instances. In the first set, the six training instances are "*good*", "*not good*", "*not not good*", "*not not not good*", "*not not not not good*" and "*not not not not not good*" with alternating labels "positive" and "negative". In the second set, the training set only consists two labeled instances, the positive phrase "*good*" and the negative phrase "*not not not not not good*". We then trained the GRU model on these two training sets, and then applied these models on a dataset by extending the first training set with two additional phrases "*not not not not not not good*" and "*not not not not not not not good*". As we can see from Figure 3, the model can infer the multiple negation correctly for the given cases when trained on the first set, and is able to generalize to unseen phrases well. However, it fails to do so for the second. This indicates that proper supervision would be needed for the models to capture the compositional nature of the semantics as conveyed by $N$-grams.

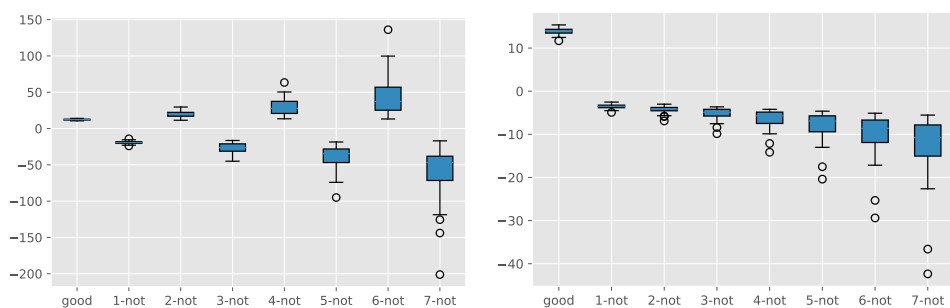

Figure 3: Distributions of polarity scores for negation $N$-grams ($N = 1 - 8$) in the phrase "*not not not not not good*". Each box represents a polarity score distribution for either the token "*good*" or the $i$ times negation $N$-grams (e.g., "3-not" refers to "*not not not good*"). Circles refer to outliers. Left, model trained on six labeled phrases. Right, model trained on two labeled phrases. Results from 30 trials with random initializations. A GRU cell is used.

## 5.2 TRAINING WITH SEQUENCE-LEVEL REPRESENTATIONS

We examined whether the sequence-level representations along with the token-level representations could capture sufficient features during training and perform on par with the standard cells. We trained models by replacing the standard GRU or LSTM cell with the corresponding $N$-gram representations ($\hat{h}_t$ for a GRU or LSTM cell). We evaluated them on both sentiment analysis and

language modeling tasks and compared them with the standard models. Additionally, we created a baseline model named "Simplified" by removing all the terms involving $x_t$ from $A(x_t)$ in Equation 11 and 16. The resulting representations do not capture sequence-level information. On the binary SST dataset (with sub-phrases) and Movie Review dataset (Pang & Lee, 2004), we found both the standard cells and our $N$-gram representations behaved similarly, as shown in Table 3. But the "Simplified" models did not perform well. Glove (Pennington et al., 2014) embeddings were used.

We ran language modeling tasks on the Penn Treebank (PTB) dataset (Marcus et al., 1993), Wikitext-2 dataset and Wikitext-103 dataset (Merity et al., 2016) respectively[3]. The embedding size and hidden size were both set as 128 for PTB and Wikitext-2, and 256 for Wikitext-103. *Adaptive softmax* (Joulin et al., 2017) was used for Wikitext-103.

It can be seen that using such representations can yield comparable results as the standard GRU or LSTM cells, as shown in Table 4. However, for the "simplified" representations, their performances dropped sharply which implies the significance of sequence-level representations. We noticed the intermediate outputs could grow to very large values on Wikitext-103, therefore, we clamped the elements in the hidden states to the range (-3, 3) at each time step. This demonstrated that the sequence-level features might be a significant contributor to the performances of a GRU or LSTM cell apart from the token-level features.

Table 3: Accuracy (%) on sentiment analysis datasets

| Cell | Type | SST | | MR | |
|---|---|---|---|---|---|
| | | Valid | Test | Valid | Test |
| LSTM | Standard | 85.8 | 87.9 | 81.6 | 78.7 |
| | $N$-gram | 86.9 | 87.9 | 82.4 | 78.4 |
| | Simplified | 83.0 | 84.8 | 81.0 | 76.2 |
| GRU | Standard | 85.9 | 88.1 | 81.2 | 76.7 |
| | $N$-gram | 86.4 | 88.3 | 82.6 | 77.6 |
| | Simplified | 82.9 | 83.7 | 80.6 | 76.1 |

Although using the $N$-gram representations perform well on both tasks, we cannot rule out the contributions of other underlying complex features possibly captured by the standard cells. This can be observed from the test perplexities obtained from the $N$-gram representations, which are generally slightly higher than those obtained from the standard GRU or LSTM cells. However, the $N$-gram representations even outperformed the standard GRU cell on Wikitext-103.

Table 4: Perplexities on language modeling datasets. All parameters were initialized randomly.

| Cell | Variants | PTB | | Wikitext-2 | | Wikitext-103 | |
|---|---|---|---|---|---|---|---|
| | | Valid | Test | Valid | Test | Valid | Test |
| GRU | Standard | 89.08 | 77.19 | 91.79 | 80.68 | 121.77 | 116.70 |
| | $N$-gram | 90.46 | 78.32 | 94.53 | 82.69 | 109.95 | 106.23 |
| | Simplified | 100.30 | 85.58 | 105.50 | 92.28 | 137.00 | 130.65 |
| LSTM | Standard | 89.22 | 77.11 | 92.23 | 80.74 | 105.25 | 101.08 |
| | $N$-gram | 90.21 | 77.82 | 94.35 | 81.21 | 107.1 | 103.34 |
| | Simplified | 101.06 | 85.64 | 105.67 | 92.13 | 135.69 | 130.07 |

# 6 CONCLUSION

In this work, we explored the underlying features captured by GRU and LSTM cells. We expanded and unrolled the internal states of a GRU or LSTM cell, and found there were special representations among the terms through a series of mathematical transformations. Theoretically, we found those representations were able to encode sequence-level features, and found their close connection with the $N$-gram information captured by classical sequence models. Empirically, we examined the use of such representations based on our finding, and showed that they can be used to construct linguistic phenomenons such as negations on sentiment analysis tasks. We also found that models using such representations only can behave similarly to the standard GRU or LSTM models on both the sentiment analysis and language modeling tasks. Our results confirm the importance of sequence-level features as captured by GRU or LSTM, but at the same time, we also note that we could not rule out the contributions of other more complex features captured by the standard models. There are some future directions that are worth exploring. One of them is to explore possible significant features captured by higher-order terms in a GRU/LSTM cell, and understand how they contribute to the performances.

---

[3]Our model is a word-level language model, we used the *torchtext* package to obtain and process the data.

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

# A APPENDIX

## A.1 DATASET STATISTICS

We listed the statistics of the datasets used in our experiments.

Table 5: Statistics of sentiment analysis datasets. "Label" refers to the size of the positive labels and negative labels in the training set, "positive/negative" for binary classification, "positive/neutral/negative" for 3-class classification.

| Data | Train/Valid/Test | Label | Vocab | Max.len | Remark |
|---|---|---|---|---|---|
| SST | 98,794/872/1,821 | 56,187 /42,607 | 17404 | 54 | sub-phrase labels |
| SST-Sampled | 320/-/- | 110/86/124 | 248 | 7 | 3-class |
| MR | 9,500/500/662 | 4,729/4,771 | 18584 | 59 | |
| Synthetic | 4,120/230/200 | 2,060/2,060 | 85 | 6 | negation cases |

Table 6: Statistics of language modeling dataset, quoted from Einstein.ai

| | PTB | | | Wikitext-2 | | | Wikitext-103 | | |
|---|---|---|---|---|---|---|---|---|---|
| | Train | Valid | Test | Train | Valid | Test | Train | Valid | Test |
| Article Num | - | - | - | 600 | 60 | 60 | 28,475 | 60 | 60 |
| Token Num | 887,521 | 70,390 | 78,669 | 2,088,628 | 217,646 | 245,569 | 103,227,021 | 217,646 | 245,569 |
| Vocab Size | 10,000 | | | 33,278 | | | 267,735 | | |

## A.2 SST DATASET

We extracted adjectives (shown in Table 7) based on their frequency ratio in the positive and negative instances. If an adjective appeared mostly in positive (negative) instances, we would regard it as a positive (negative) adjective. The *textblob* package [4] was used to detect adjectives. The labeled phrases (shown in Table 8) would be selected from the SST dataset if they had labeled sub-phrases of opposite signs by removing the negation words.

Table 7: Extracted adjectives from the SST dataset

| Type | Adjectives | Size |
|---|---|---|
| Positive | outstanding, ecological, inventive, comfortable, nice, authentic, spontaneous, sympathetic, lovable, unadulterated, controversial, suitable, grand, happy, enthusiastic, adventurous, successful, noble, true, detailed, sophisticated, sensational, exotic, fantastic, remarkable, impressive, charismatic, good, effective, rich, popular, unforgettable, famous, comical, energetic, ingenious, extraordinary, pleased, tremendous, marvelous, believable, artistic, expressive, exceptional, fabulous, strong, humorous, gorgeous, available, incredible, memorable, likable, delicious, attractive, impeccable, accessible, delighted, hilarious, inspirational, thoughtful, affable, creative, great, imaginative, enjoyable | 65 |
| Negative | bad, tedious, miserable, psychotic, didactic, inexplicable, feeble, sloppy, disastrous, stupid, amateurish, false, cynical, farcical, terrible, unhappy, horrible, atrocious, idiotic, wrong, pathetic, angry, uninspired, vicious, unfocused, unnecessary, artificial, troubled, questionable, arduous, stereotypical, poor, unpleasant, forgettable, ridiculous, laughable, indecipherable, grievous, impassive, gratuitous, weak, simplistic | 42 |

### A.2.1 BIDIRECTIONAL GRU/LSTM MODELS

We also conducted experiments using bidirectional GRU and LSTM cells on the binary SST dataset. Figures 4 and 5 show that the models can capture such negation from both directions. For example, the four-gram "to feel contradictory things" has a negative sequence-level polarity score while the five-gram "freedom to feel contradictory things" has a positive one. Similarly, in the backward direction, the bigram "to freedom" has a positive sequence-level polarity score while the five-gram "things contradictory feel to freedom" has a negative one. Glove (Pennington et al., 2014) embeddings were used. Embedding size and hidden size were set as 300 and 1024 respectively.

---

[4]https://textblob.readthedocs.io/en/dev/

Table 8: Selected labeled phrases from the SST dataset

| Type | Phrase |
|---|---|
| positive | never fails him, never fails to fascinate ., not a bad way, never wanted to leave ., never feels derivative, not to be dismissed, never becomes claustrophobic, never fails to entertain, never feels draggy, never veers from its comic course, never growing old, not a bad premise, not without merit, not mean - spirited |
| negative | not life - affirming, not for every taste, hardly an objective documentary, not a must - own, never takes hold ., not a classic, not the best herzog, never rises to a higher level, not exactly assured in its execution, not as good as the original, not be a breakthrough in filmmaking, not always for the better, not well - acted, not very compelling or much fun, never reach satisfying conclusions, not as sharp, never rises above superficiality ., never seems fresh and vital ., not a good movie, not a movie make, not the great american comedy, not smart and, hardly seems worth the effort ., not enough of interest onscreen, not funny performers, not well enough, not in a good way, hardly a nuanced portrait, not good enough, not number 1, not very amusing, never gaining much momentum, not well enough, not one clever line, not a good movie, never comes together |

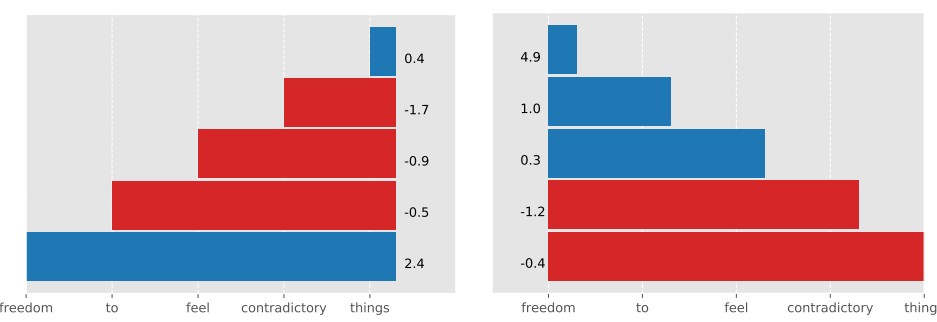

Figure 4: Polarity scores for $N$-grams ($N = 1 - 5$) in the phrase "freedom to feel contradictory things" from bidirectional GRU cells. Left, forward cell; right, backward cell. Each bar represents a $N$-gram. Red refers to negative polarity scores while blue refers to positive ones.

### A.2.2  3-CLASS SST DATASET

We also considered the dissenting scenarios for 3-class sentiment analysis. We extracted 160 pairs of labeled phrases starting with negation words and their sub-phrases with different labels from the 3-class SST dataset. We trained the model on the extracted pairs directly until all the instances were classified correctly. Table 9 showed that the sequence-level polarity scores of the longest $N$-grams in the phrases could generally capture the differences between the pairs, and dominate in the dimensions corresponding to the labels.

Table 9: Results on the sampled 3-class SST dataset. "neu", "pos" and "neg" refer to that the sequence-level polarity scores have the largest value in the dimension corresponding to the label "neutral", "positive" and "negative".

| Type | LSTM | | | GRU | | | Num | Example |
|---|---|---|---|---|---|---|---|---|
| | neu | pos | neg | neu | pos | neg | | |
| Neutral $N$-gram | 30 | 2 | 5 | 37 | 0 | 0 | 37 | not ultimate blame |
| Positive $N$-gram | 0 | 23 | 5 | 6 | 18 | 4 | 28 | never feels derivative |
| Negative $N$-gram | 1 | 2 | 92 | 6 | 0 | 89 | 95 | not to see it |

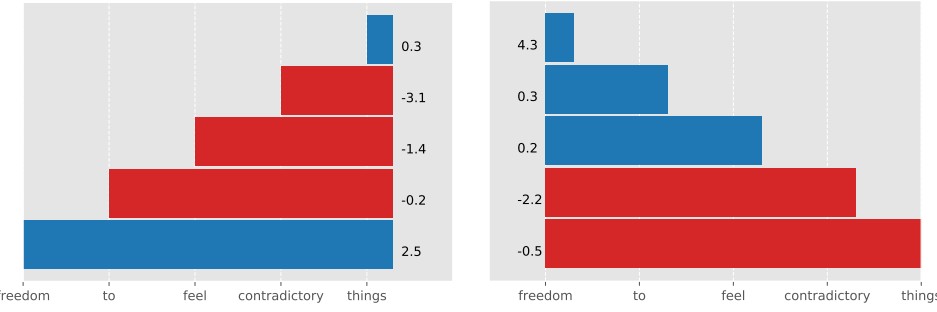

Figure 5: Polarity scores for $N$-grams ($N = 1 - 5$) in the phrase "freedom to feel contradictory things" from bidirectional LSTM cells. Each bar represents a $N$-gram. Red refers to negative polarity scores while blue refers to positive ones.

### A.2.3   IMPACT OF $N$-GRAM LENGTHS

To understand the impact of sequences with different lengths, we selected positive and negative $N$-grams ($N$=1-4) ending with the last token of the instances based on their association with positive and negative labels respectively. It appears that the token-level polarity scores for unigrams and sequence-level polarity scores for bigrams can reflect their association with labels better than that for trigrams and four-grams as shown in Figure 6. This demonstrates that although the sequence-level features can be well captured and are important for sentiment analysis tasks, the shorter sequences in general may be playing more crucial roles than the longer ones.

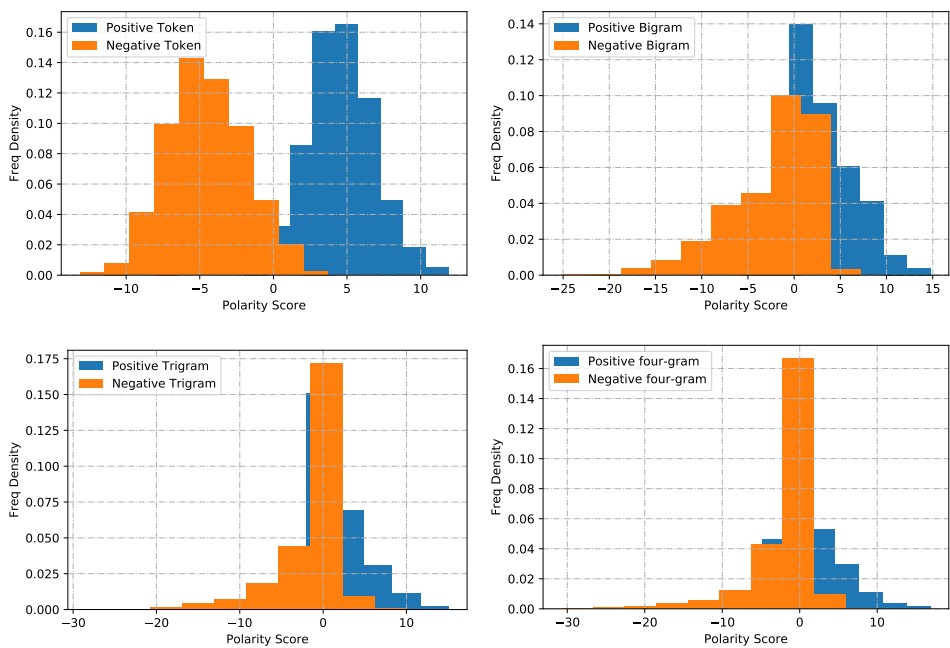

Figure 6: Polarity score distribution for the $N$-grams ($N$=1-4) that have strong association with a specific label. Result from a GRU cell on the SST dataset.

### A.3   SYNTHETIC DATASET

Few real-world datasets provide sufficient exposures for tokens, phrases and their negation expressions. Therefore, we synthesized instances with binary labels based on adjectives and negation

Table 10: Polarity score distribution for tokens, negation phrases and double negation phrases. Results from 10 trials with random initializations.

| Type | | Number | GRU Polarity Score | LSTM Polarity Score |
|---|---|---|---|---|
| positive adjectives | words | 15 | $4.06 \pm 0.62$ | $3.91 \pm 0.86$ |
| | negation | 31 | $-5.85 \pm 4.81$ | $-14.36 \pm 8.32$ |
| | double negation | 10 | $12.75 \pm 5.66$ | $21.17 \pm 21.00$ |
| negative adjectives | tokens | 18 | $-4.17 \pm 0.75$ | $-3.75 \pm 0.90$ |
| | negation | 37 | $6.59 \pm 5.11$ | $15.05 \pm 9.61$ |
| | double negation | 10 | $-11.27 \pm 8.08$ | $-22.13 \pm 18.50$ |

Table 11: Adjectives, negation and double negation examples for the synthetic dataset

| | Adjectives | Negation Example | Double Negation Example |
|---|---|---|---|
| Positive | good, nice, charming, awesome, fascinating, impressive, excellent, wonderful, attractive, interesting, inspiring, stunning, amazing, terrific, incredible | not charming, not good
not incredible, not wonderful
not nice, not amazing
never charming, never amazing
never incredible, never attractive
never look wonderful, never look charming
never seemed good | not not attractive
not not inspiring
not not awesome
never never awesome
never never wonderful
never never seemed incredible |
| Negative | awful, bad, uninspiring, dull, boring, tedious, horrible, terrible, pathetic, mediocre, shallow, pointless, unfunny, gross, poor, dreadful, dire, useless | not horrible, not pointless
not dire, not mediocre
not terrible, not unfunny
not bad, not dull
never boring, never horrible
never seem terrible, never seem pointless
never seemed unfunny | not not gross
not not mediocre
not not unfunny
never never bad
never never seemed dire
never never seemed shallow |

words following simple grammatical rules. We created a vocabulary of 85 words including nouns, verbs, adjectives, adverbs, articles and negation words. Negation phrases and double negation phrases are also incorporated in those instances. We let positive adjectives such as "inspiring" and the double negation expression "not not inspiring" appear in the instances labeled as positive, e.g., "her dramas were indeed inspiring", "her movies are not not inspiring" .Let the negation expression "not inspiring" appear in the instances labeled as negative, e.g., "his movie is not inspiring". We did similarly for negative adjectives.

Table 10 shows that the sequence-level representations could generally have polarity scores matching the roles of the negation and double negation $N$-grams. For the positive (negative) adjectives, their negation $N$-grams have negative (positive) sequence-level polarity scores while their double negation $N$-grams have positive (negative) ones. The negation tokens are "*not*" and "*never*". Models were trained until all the negation expressions have been classified correctly.

The key tokens, negation and double negation phrases are shown in Table 11.

### A.4 PERFORMANCE DURING TRAINING

We make comparisons between the performances of the standard GRU/LSTM cell and the $N$-gram representations during training. It can be seen from Figure 7 and 8 that the $N$-gram representations perform similarly to the standard GRU/LSTM cell on the PTB, Wikitext-2 and Wikitext-103 datasets. Adam optimizers (Kingma & Ba, 2014) were used.

### A.5 MULTIPLE NEGATION ON A SIMPLE SET

To scrutinize the sequence-level features under controlled conditions, we created a training set consisting of the phrases: "good", "not good", "not not good", "not not not good", "not not not not good" and "not not not not not good" with alternating labels "positive" and "negative". We trained a standard GRU or LSTM cell on the training set until the loss converged (less than $10e$-6) with the embedding size 256, hidden size 1024, then calculated the corresponding polarity scores for $N$-grams.

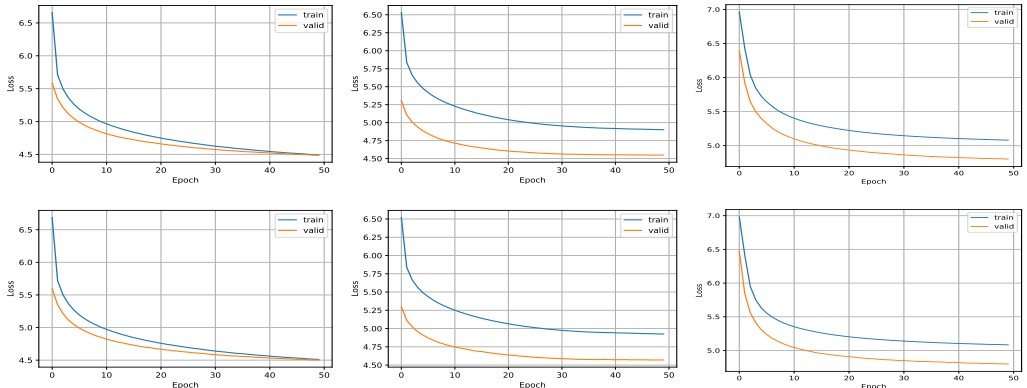

Figure 7: Top, standard GRU Cell; bottom, approximate hidden state representation. From left to right: PTB, Wikitext-2, Wikitext-103. Training and validation losses for language modeling tasks.

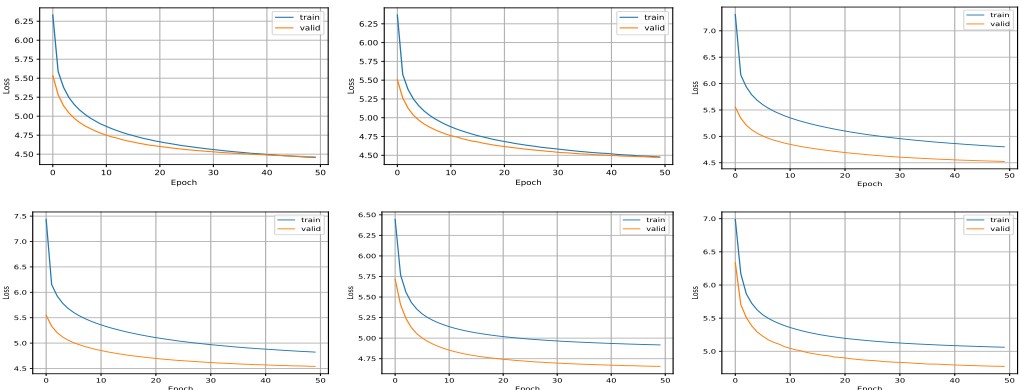

Figure 8: Top, standard LSTM Cell; bottom, approximate hidden state representation. From left to right: PTB, Wikitext-2, Wikitext-103. Training and validation losses for language modeling tasks.

Figure 9 shows the sequence-level features derived from the pre-trained GRU or LSTM cell were able to detect multiple negation, implying those features were likely to be significant for classification decisions. For example, the four-gram "not not not good" generally has a large negative sequence-level polarity score while the five-gram "not not not not good" generally has a large positive one, and the six-gram "not not not not not good" has a large negative one again. These polarity scores could help the phrases reverse the polarity of their sub-phrases with opposite labels. Figure 10 shows examples from a pre-trained GRU and an LSTM cell with random initializations. We can see with each "*not*", the sequence-level polarity score will reverse the polarity. Based on aforementioned analysis, it may be essential for the models to have enough exposure to relevant structures during the training phase.

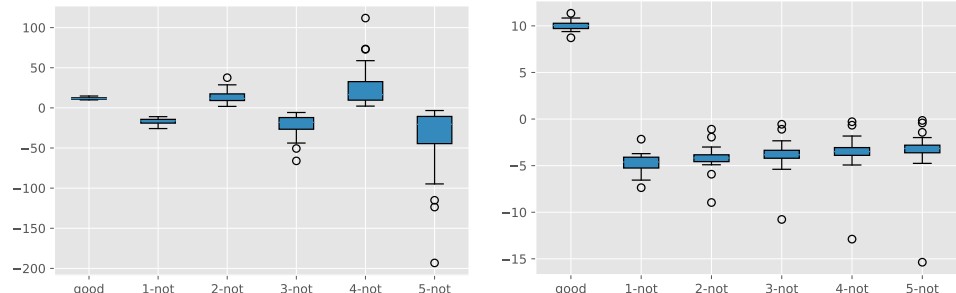

Figure 9: Distributions of polarity scores for negation $N$-grams ($N = 1 - 6$). Each box represents a polarity score distribution for either the token "good" or the $i$ times negation $N$-grams (shown as $i$-not, $i = 1, 2...5$). Circles refer to outliers. Results from 30 trials with random initializations. An LSTM cell is used.

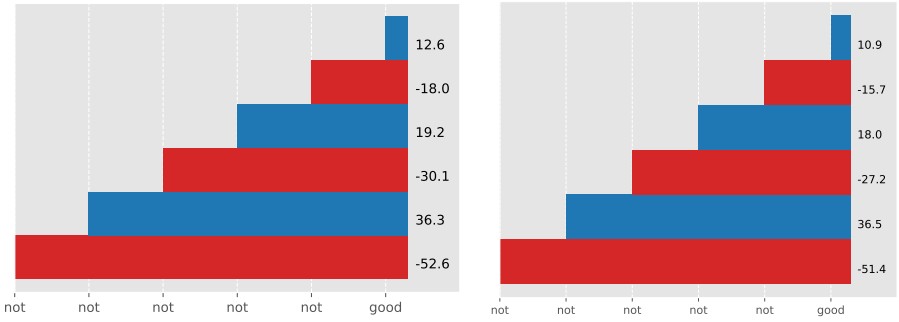

Figure 10: Polarity scores for $N$-grams ($N = 1 - 6$) in "not not not not not good". Left, GRU; right, LSTM. Each vertical bar represents the polarity score (token-level or sequence-level) for the $N$-gram that it covers. Red bars refer to negative polarity scores and blue bars refer to positive scores.

