# OpenReview forum: "SEQUENCE-LEVEL FEATURES: HOW GRU AND LSTM CELLS CAPTURE N-GRAMS"
_ICLR.cc/2021/Conference — Reject_

### Official Review · AnonReviewer1 · 2020-10-18
**Linear LSTMs**

**Rating:** 4
**Confidence:** 2

**Review:**

DISCLAIMER: this is not my field of research. With strong arguments I could be persuaded to change my score.

This paper introduces a method to unroll the Gated Recurrent Unit (GRU) and the Long Short-Term Memory (LSTM) unit using a taylor expansion. Essentially a linearization of the GRU and LSTM. Given the simplifications, the authors argue that their model captures N-gram information for the sequential information. The paper presents results suggesting that this approximation is able to capture much of the same sequential information as the LSTM and GRU on benchmarks such as SST-2, PDB, Wikitext-2, and Wikitext-103.

What this paper excels at is a thorough theoretical formulation of the proposed approximation, and a comparison with an approximation without the sequential information. This shows that the approximate version is able to capture sequential information.

However, the relevance of this paper is not clear to me, and the introduction and related works does little more to explain that relevance than stating: “understanding the essential features captured by GRU/LSTM”. The tasks that the method is tested on are synthetic data for sentiment and tasks/models that haven't been relevant since 2016. The motivation for these tasks, and the qualitative analysis is hard for me to understand.

The paper could use some reformulations and more emphasis on what exactly the purpose of the paper is, in particular I find the lack of consistency in present/past tense disrupts the reading experience.

Below I have made a few comments/questions:

Abstract:
“Sequential data in practice” - unclear what this means
“sequence-level representations brought in by the gating mechanism” - I dont understand this sentence
“essential components” - vaquely defined
“Based on such a finding” - rephrase

Introduction:
“gradient vanishing or explosion issues” - vanishing or exploding gradients
“While such models ...” - this whole sentence is a little vague

Related work:
“With the variants” - its variants
General comment (also for introduction): While you mention many interesting findings in recent years, it is difficult for me to assess how exactly your work differs. Please use the related works to emphasize what you are doing differently than previous work in your field.

LSTM:
“A LSTM cell” - An LSTM cell

Experiments:
“Figure 2” I don’t get what each bar represents
“Subphrase labels” - are subphrase labels the node annotation?

Why is negation, and a synthetic variant, important to explain the relationship between N-grams and LSTMs?
Why do you choose the datasets you do? Why is SST-2 and benchmarking against older language models of interest? Why is an N-gram comparison interesting? Perhaps the authors should look into contemporary research on formal methods in sequential models for inspiration of tasks and where an interesting hypothesis might be: https://arxiv.org/abs/1906.03648

Update:
I have read the rebuttal and the updated paper. I don't see my issue of relevance addressed. My score remains the same.

---

> ### Author Response · Authors · 2020-11-25
> **Thanks for your thoughtful feedback.**
>
> “sequential data in practice” refers to the data in sequences, for example, a sentence consisting of a sequence of words. Changing the order of the words will probably result in a different sentence.
> “essential components” refers to those sequence-level features.
> “sequence-level representation” refers to the term $A(x_t)A(x_{t-1})...A(x_{i+1})f(x_i)$
> that could capture features of a sequence ranging from $x_i$ to $x_t$. Each bar in
> Figure 2 (in the original version) represented a $N$-gram starting from the left side and
> ending with the right side.
> “sub -phrases” means the SST dataset has labels for sub-phrases.
>
> WeI have revised the background part and highlighted the differences between previous works and my work.
>
> Negation cases can be one type of testbed to examine the sequence-level features. If those features are significant, they will have corresponding polarity scores in a desired way for decisions on negation cases. For example, if “good” is labeled as positive and “not good” is labeled as positive, the sequence-level polarity score for the bigram “not good” will likely be negative to reverse the polarity. We observed that in our experiments. We also considered 3-class sentiment analysis in which sequence-level features could likely capture the label differences. The recommended paper seems useful to us. We will follow the work in the future.
>
> There will not be clear patterns for  sequence-level polarity scores if those negation expressions involved are not exposed to sufficient labeling information. For example, if we only have labeled phrases “not not not good” and “good” in the training set but without the label information for the trigram  “not  not good”, the model will not necessarily learn a clear pattern for  “not  not good”.  We have added such experiments in the revised version. Therefore, we created synthetic datasets that provide sufficient exposure for those negation cases.

---

### Official Review · AnonReviewer3 · 2020-10-28
**Solid reasoning and Interesting Results**

**Rating:** 6
**Confidence:** 4

**Review:**

This paper provides a reliable interpretation of modern RNN models, through unrolling GRU and LSTM cells. The approximate state representations include a token-level term that only depends on the current input token and a sentence-level term that depends on all inputs until the current token. The deriving process is clear and illuminating. The experiment section shows that the approximation shares similar behavior and performance as the original model.

Overall, the paper is well written and easy to follow. Although GRU and LSTM are no longer the default model for SOTA performance in the NLP community. I believe that this study still provides interesting insights for those who want to develop better recurrent or non-recurrent models in the future.

My major concern is that the language model experiment didn’t include a stronger baseline method, such as AWD-LSTM, which provides a significantly lower ppl compared to these in the paper.

It would be interesting to see a more detailed ablation study, that studies the importance of each term in A(x_t).

---

> ### Author Response · Authors · 2020-11-25
> **Thanks very much for your helpful comments.**
>
>
>
> We did not seek to prove that those representations could achieve state-of-the-art performances here. Designing new architectures based on the findings could be one of the future directions.
>
> Our focus is to explore and study significant features captured by GRU or LSTM cells. We conducted experiments on language modeling tasks to assess the significance of the sequence-level and token-level features. If such features are significant for standard cells, they will produce comparable performances without other possible features.
>
> We did ablation studies by removing terms in $A(x_t)$. We removed the terms such as  $diag(x^n_t )W_{hz}$ which lead to a drop of the performance. We would do more systematic work in the future.

---

### Official Review · AnonReviewer4 · 2020-10-29
**An interesting attempt to improve theoretical understanding on gated RNNs.**

**Rating:** 5
**Confidence:** 4

**Review:**

This paper attempts to add a contribution on understanding how gated recurrent neural networks like GRUs and LSTMs can learn the representation of n-grams. The authors expand the sigmoid function and the hyperbolic tangent function using Taylor series to obtain approximated closed-form mathematical expression of hidden representation when using the GRU or the LSTM as the update rules. The approximated hidden representation of the GRU and the LSTM update rules can be separated into two terms, (1) the current token-level input feature and (2) the sequence-level feature, which is a weighted sum of all previous tokens. As the hidden representation consists of two feature terms, one can take each feature (either token-level or sequence-level) separately for a downstream task, e.g., evaluate how good when sequence-level feature is used for predicting polarity score in sentiment analysis.

The idea of improving theoretical understanding on how n-grams are modelled by gated recurrent activation functions is sound. However, I am not entirely satisfied with what has been investigated after obtaining the approximated closed-form expression of gated recurrent activation functions. The tasks that were used in the experiments are sentiment analysis and language modelling. In sentiment analysis, most of the plots were there to show how token-level features or sequence-level features align with the polarity score, and we can observe some sort of individual implication from each term. However, it is predictable that sequence-level feature should be meaningful. I don't see much of insights by showing that the polarity score from sequence-level features indeed align with this prediction. If we can to apply Taylor expansion to simple recurrent neural networks (RNNs), such that we can expand the hidden representation of a standard RNN into two terms: the current token-level input feature and sequence-level feature, how would the results look like and how can we relate them with what were reported in this paper? Is this paper particularly showing how gated RNNs are modelling n-grams or RNNs in general? A comparison would be nice to show how sequence features get improved in gated RNNs.

It is interesting to see that the approximated versions of GRUs and LSTMs can perform on a par with the original models on language modelling tasks, however, these results don't necessarily improve our understanding on how gated RNNs are capable of learning good representations of n-grams. They confirm that sequence features are indeed helpful though.

In Section 5.1, if there were multiple trials of experiments on the same task, why not report the average and the variance of the results instead of one set out of multiple results?

In Section 5.2, Adpative softmax (Joulin et al., 2017) was used for Wikitext-130. -> Adpative softmax (Joulin et al., 2017) was used for
Wikitext-103.

---

> ### Author Response · Authors · 2020-11-25
> **Thanks very much for your constructive feedback.**
>
>
> We aimed to explore and study significant features captured by GRU or LSTM cells through mathematical transformation. We found there were matrix-vector multiplications that could capture sequence-level information in the expanded hidden states. Perhaps this could give us some inspiration for the gating mechanism and the internal mechanism of GRUs or LSTMs.
> We have also discussed vanilla RNN cells and made comparisons with GRU cells and LSTM cells in the revised version
> Moreover, we have revised the experiment part and reported more results from multiple trials.

---

### Official Review · AnonReviewer2 · 2020-10-30
**nice idea but makes a lot of questionable assumptions**

**Rating:** 3
**Confidence:** 3

**Review:**

This paper examines n-gram level features encoded within the hidden states of recurrent neural models. The proposed method approximates the hidden states of LSTMs and GRUs with a first-order Taylor series, which the authors claim is an adequate approximation with small enough inputs. The authors apply their method to models trained on synthetic sentiment analysis and language modeling datasets. The paper is difficult to understand, and many assumptions are not properly justified. The experiments are also not convincing, as a large portion of the analysis focuses on small synthetic data, and some of them do not have clear takeaways or explanations as to why they were conducted in the first place. Overall, I cannot recommend the paper's acceptance in its current form.

comments/questions:
- is the scenario of extremely low input magnitudes realistic? how generalizable are these findings to standard initializations used in NLP architectures
-  similarly, on page 3 the authors assume that the higher order terms of h_{t-1} are "insignificant"; in practice, it is unclear how often this is true. i wish the paper would contain more justification behind these assumptions, as they are critical for judging how faithful the approximation is and thus how useful the proposed method is for diagnosing RNNs.
- can't the authors actually show quantitatively how good the approximations are? if the higher-order terms indeed do not affect the quality of the approximation, that could be justified by some experiments. i'm not really convinced by Table 2: the approximations could be quite different from the original model but still yield good downstream accuracy.
- what is a "polarity score" (bottom of page 5)? I didn't quite understand what this is supposed to represent, is it how predictive of a label a particular span is?
- why are synthetic datasets used at all here? experiments on a small set of sentences with a tiny vocabulary and artificial "double negatives" are not compelling. Appendix A.2 does not fully specify this dataset (nor motivate why it was created); what is e.g., its average sentence length?
- the results on a real sentiment dataset (SST2) are confusing (sec 5.1.3): what does figure 4 show me that I couldn't already learn by simply passing those two phrases into the model as separate inputs and looking at the model's prediction?
- what is the point of training models with the approximations instead of the original GRU/LSTM cell equations?  i don't understand the significance of Table 3.

---

> ### Author Response · Authors · 2020-11-25
> **Thanks very much for your thoughtful comments.**
>
> Our purpose was not to approximate the output of a standard GRU or LSTM cell precisely, but to locate significant features they could possibly capture. We conducted experiments with as-is initializations on both synthetic and real-world datasets and found the features were likely to reflect properties as expected.
>
> We have modified the assumption in our revised version. As we have mentioned in the original version, we could not rule out the influence of those high-order terms, but we chose to focus on the sequence-level features and assess their significance.
> The gap between the hidden states of a standard cell (either GRU or LSTM) and the approximated one could be obvious in real scenarios. But the sequence-level features can be significant for the decisions and our experiments supported that.
> A polarity score is a metric that measures how strong a token or a sequence is associated with a specific label. For example, if the token “good” is strongly associated with the “positive” label, it will likely have a large positive polarity score in the end and make a significant contribution to decisions.
>
> There will not be clear patterns for sequence-level polarity scores if those negation expressions involved are not exposed to sufficient labeling information. For example, if we only have labeled phrases “not not not good” and “good” in the training set but without the label information for the trigram  “not  not good”, the model may not be able towill not necessarily learn a clear pattern for  “not  not good”.  We have added such experiments in the revised version. Therefore, we created synthetic datasets that provide sufficient exposure for those negation cases.
>
> Figure 4 (in the original version)  indicated the features can capture negations from both-sides.  We have revised this part.
> We would like to examine whether the  sequence-level representations along with token-level representations could capture significant features during training. If they were the major contributors to the performances of GRU or LSTM cells,  they could work as well as the standard cells during training even without other representations in the expanded GRU and LSTM hidden states. Our experiments showed they could produce comparable performances on the sentiment analysis and language modeling datasets.

---

### Official Review · AnonReviewer5 · 2020-11-05
**Linearizing GRUs and LSTMs allows for decomposing into unigram- and substring contributions and that does well enough on sentiment analysis**

**Rating:** 4
**Confidence:** 4

**Review:**

This paper proposes to linearize GRU and LSTM cells (as error terms should be negligible when inputs are small in magnitude). Putting these linearized, or, really, affine, RNN cells together into a single-layer sequence processor, thanks to the affine-ness, we can decompose the score that is obtained by taking dot products with a query at each timestep into contribution by immediate unigram features and all subsequences leading to this unigram. The authors evaluate these scores, showing that they do and don't capture phenomena in a synthetic dataset and proceeding to show that when both training and evaluating with this simplified network on SST yields strong results.

#### Strengths/what I loved:

- Motivation and Related Work seemed nicely done, set this paper up nicely, and made me excited to read on!
- I like the idea of testing double negation and omitting it during training and it was interesting to see the networks then fail to pick up on it (assuming the synthetic dataset is reasonable).
- The visualizations of sequence-level scores (Figures 2 and 4) are very cleverly chosen and powerful, even if they take a bit of getting used to.
- Figures 3 and 5 are striking: even on SST unigrams and perhaps bigrams seem to get you most of the way if you trust the approximate interpretation.

#### Criticism/weaknesses:

- It is unclear how the simplified/affine-ized architecture relates to vanilla RNNs---those, too, can be decomposed like that and one could just as well look at these features. To paraphrase: I don't see why any of the analysis and results in this paper is only true for gated cells and I wouldn't be surprised to see a vanilla RNN yield equally good approximate cells, since after all the task is very simplistic (Section 5.2 concludes as much, as even without sequence information the task is easy to solve, but the conclusion drawn from that result, namely that there is something about GRUs and LSTMs to read from this doesn't follow in my opinion).
- The synthetic dataset is a mystery: yes, it contains sentences that contain the words shown in Table 6 (Appendix A.2), but... what are these sentences? Are they actual text from some dataset? Text sampled from some model or grammar? Random words without any sequential coherence? What is the vocabulary size? What does appearing "mostly" in positive or negative instances really mean? Just giving one or two examples would have made me a lot less worried and confused about what is going on here, but to base many if not most of the results on it, this dataset is woefully underdescribed.

#### Questions:

- Notation/definitions in section 5.1.1 are either very unclear or flat-out wrong: Sun & Lu (2020) do indeed define a notion of a "token-level polarity score" for each output class, but the notion of output classes is not mentioned at all here, in fact instead of the output embedding matrix W that Sun & Lu (2020) posit, this paper speaks only of a single vector w. I assume that that is the vector associated with the positive class and the prediction thus is strictly binary---very much unlike Sun & Lu (2020). In addition to that, the notion of a "sequence-level polarity score" that here complements the token-level score is not at all mentioned in Sun & Lu (2020) as far as I can see, so to say that their methodology is used is misleading in more than one way. Finally, the model described in Sun & Lu (2020) is one that uses attention, i.e., that does mean-pooling after the linear layer that is transforming the individual states. This paper mentions the linear layer, but not the pooling/attention, so it's unclear if that is a poor paraphrase of Sun & Lu or whether this paper here too diverges from the paper it claims to build on.
- Are Figures 2 and 4 cherry-picked? Since nothing else is stated, I would assume so. In light of that, the longest subsequence on the right of Figure 2 being negative is rather strange. Do you have an explanation? Is "those" a negative word?
- I think it would've been very interesting to see whether or not this simplification is legitimate, that is whether the assumptions made in the derivation are justified: train with the original cell formulation, but then evaluate *quantitatively* using the approximate cell to essentially create a table like Table 2 or perhaps even a scatter plot for individual logits to see how much things change or don't change. That would go a long way to convince me that the approximation is at least reasonable.
- A.3: "there's an apparent difference" What is that difference? I don't see any.
- A.4: What is the change here, can you highlight it or motivate it? The results certainly aren't particularly impressive, so I'm tempted to say this section hurts more than it helps...

#### Typos and other small things:

- 5.1: As you have multiple runs, which one was selected? The best according to some metric? The median or mean somehow? Or randomly chosen? Either way, I strongly feel that empirical results should always come with a sense of stability: what was the variance between runs? Between hyperparameters? How sensitive are results and what can we say about statistical significance?
- The overlaid histograms definitely need to have some transparency or be shown another way---right now it is impossible to see what is "happening" in the blue bars as they are hidden by the orange bars.
- The last sentence of Section 5 should link to A.4, I guess?


---

I read and appreciated the response, but my overall rating is still leaning negative.

---

> ### Author Response · Authors · 2020-11-25
> **Thanks very much for your helpful comments.**
>
> Relation to vanilla RNNs. The unrolled architecture of either a GRU cell or an LSTM cell is different from vanilla RNN cells. In the paper, we extracted the first two parts from the unrolled architecture of a standard GRU cell or a standard LSTM cell and represented them as: $hat h_t = f(x_t) + \sum_{i=1}^t A(x_t)...A(x_{i+1}) f(x_i)$,  which consists of two parts: token-level feature and sequence-level features.
> We can also linearize a vanilla RNN as: $hat h_t = f(x_t) + \sum_{i=1}^{t-1} W^{t-i} f(x_i)$. Each term $W^{t-i} f(x_i)$ in the right part cannot be viewed as a sequence-level feature as it does not contain the input information other than x_i.  The matrix-vector product $A(x_t)...A(x_{i+1}) f(x_i)$ can be more expressive than  $W^{t-i} f(x_i)$. Thanks for highlighting this question and we have revised the paper accordingly to discuss this finding.
>
> The synthetic dataset. The synthetic sentences were created based on simple grammatical rules. Our purpose was to create a balanced training set that provides sufficient labels for polarity tokens and their negated expressions. Then we can examine whether models can capture negation and double negation with respect to the sequence-level features for both positive and negative words. We have provided more details in the appendix of the revised version.
>
> Questions:
> 1.We adopted the idea of polarity score from the work of Sun & Lu (2020)  to measure the association between a token (or a sequence) and a specific label.  For 3-class (or more classes) sentiment analysis, the fully-connected layer weight is a matrix, and the output overall polarity score is a vector corresponding to label space. Actually, we can also do so for binary classification and the output polarity score will be 2-dimension. But very often for binary classification, the fully-connected layer weight is a vector, and the output overall polarity score is a scalar which will be used for making classification decisions, e.g., positive instance if it is greater than 0, negative instance otherwise.  We can also define such polarity scores for our sentiment analysis model based on RNNs. Based on our analysis, if we use the final hidden state for classification, the overall polarity score of a phrase will be viewed as the sum of the token-level polarity score  (last token), all the sequence-level polarity scores and the scores produced by other terms.   If they are significant, they will be expected to behave in the desired way on downstream tasks.
>
> 2.We ran multiple trials for each model, and we found the patterns were similar. The figures were reported based on results from random initializations. We have reported more results from multiple trials in our revised version. The sequence-level features will develop apparent patterns under sufficient exposures to certain labeling information. For example, if we only have labeled phrases “not not not good” and “good” in the training set but without the label information about “not not good”, the model may fail to learn the semantics of “not” and hence may not be able to correctly label the phrase “not not good” the model will not necessarily learn a clear pattern for  “not not good”. We have added such experiments in the revised version.  Similarly, there are no specific patterns for the polarity scores of the entire sequence “those filmmakings were not not charming”, and we observed it was close to 0 in another trial. But the patterns for the trigram “not not charming” and “not charming” are clear.
>
>  3.We did not seek to approximate the hidden states precisely. The gaps might be large if the input magnitudes were large. We believe that the sequence-level and token-level features are significant among all the possible features captured by GRU or LSTM cells.  We examined so on sentiment analysis tasks as well as language modeling tasks.
>
> 4.We have revised the sentence.
>
> 5.Certain coefficients and constants were changed compared to the original approximate hidden states. We have removed the part for proposed architectures and focused on examining sequence-level features.

---

### Decision · Program_Chairs · 2021-01-07
**Final Decision**

**Decision:**

Reject

**Comment:**

the authors demonstrated that vanilla RNN, GRU and LSTM compute at each timestep a hidden state which is the sum of the current input and the weighted sum of the previous hidden states (weights can be either unit or complicated functions), when sigmoid and tanh functions are replaced by their second-order taylor series each. they refer to the first term as token-level and the second term as sequence-level, and claim that the latter can be thought of as summing n-gram features in the case of GRU & LSTM due to the complicated weight matrices used for the weighted sum, largely arising from the gating mechanisms.

the reviewers are largely unsure about the significance of the findings in this paper due to a couple of reasons with which i agree. first, it is unclear whether the proposed approximation scheme is enough to capture much of what happens within either GRU or LSTM. if we consider a single step, it's likely fine to ignore the O(x^3) term arising from either sigmoid or tanh, but when unrolled over time, it's unclear whether these error terms will accumulate or cancel each other. without either empirically or theoretically verifying the sanity of this approximation, it's difficult to judge whether the authors' findings are specific to this approximation scheme or do indeed reflect what happens within GRU/LSTM.

second, because the authors have used relative simple benchmarks to demonstrate their points, it is difficult, if not impossible, to tell whether the authors' findings are about the datasets themselves (which are all well known to be easily solvable or solvable very well with n-gram classification models and n-gram language models) or about GRU/LSTM, which is related to the first weakness shared by the reviewer. the observations that n-gram models and simplified GRU/LSTM models work as well as the original GRU/LSTM models on these datasets might simply imply that these datasets don't require any complicated interaction among the tokens beyond counting n-grams, which lead to the original GRU/LSTM trained to be simplified (n-gram detectors.)

that said, i still believe this direction is important and is filled with many interesting observations to be made. i suggest the authors (1) verify the efficacy of their approximation scheme (probably empirical validation is enough, and (2) demonstrate their point with more sophisticated problems (carefully designed synthetic datasets are perfectly fine.)